# Amortization Transformer for Brain Effective Connectivity Estimation from fMRI Data

**DOI:** 10.3390/brainsci13070995

**Published:** 2023-06-25

**Authors:** Zuozhen Zhang, Ziqi Zhang, Junzhong Ji, Jinduo Liu

**Affiliations:** The Beijing Municipal Key Laboratory of Multimedia and Intelligent Software Technology, Beijing Institute of Artificial Intelligence, Faculty of Information Technology, Beijing University of Technology, Beijing 100124, China; zzz3582@emails.bjut.edu.cn (Z.Z.); ziqiz447@gmail.com (Z.Z.); jjz01@bjut.edu.cn (J.J.)

**Keywords:** amortization learning, transformer, brain effective connectivity, functional magnetic resonance imaging

## Abstract

Using machine learning methods to estimate brain effective connectivity networks from functional magnetic resonance imaging (fMRI) data has garnered significant attention in the fields of neuroinformatics and bioinformatics. However, existing methods usually require retraining the model for each subject, which ignores the knowledge shared across subjects. In this paper, we propose a novel framework for estimating effective connectivity based on an amortization transformer, named AT-EC. In detail, AT-EC first employs an amortization transformer to model the dynamics of fMRI time series and infer brain effective connectivity across different subjects, which can train an amortized model that leverages the shared knowledge from different subjects. Then, an assisted learning mechanism based on functional connectivity is designed to assist the estimation of the brain effective connectivity network. Experimental results on both simulated and real-world data demonstrate the efficacy of our method.

## 1. Introduction

Recently, there has been an increasing interest in brain functional network analysis. A brain functional network can be represented as a graph structure naturally, which consists of brain regions as nodes and connections among brain regions as edges [1]. According to the characteristics of functional integration and separation in the brain, brain connectivity is typically divided into two categories: functional connectivity (FC), and effective connectivity (EC).

With the development of functional magnetic resonance imaging (fMRI) technology, it has become a valuable tool for studying the workings of the human brain and has been used to investigate a wide range of cognitive processes. fMRI is a non-invasive neuroimaging technique to measure changes in blood flow and oxygenation levels in the brain that allows researchers to visualize brain activity in response to various stimuli or tasks. Both functional and effective connectivity can be generated from functional magnetic resonance imaging (fMRI) data, FC describes statistical dependencies between brain regions, while EC is the causal influence that one brain region exerts over another [2,3], which characterizes the causality between brain regions. As different individuals exhibit different EC networks, the discrepancies among EC networks offer an effective way to evaluate normal brain functions and the brain injuries related to some neurodegenerative diseases [1], such as Alzheimer’s disease [4], Parkinson’s disease [5] and autism spectrum disorder [6]. Therefore, the estimation of brain effective connectivity from fMRI data is a critical scientific problem in the investigation of the human brain connectome.

Over the past few years, there has been a discernible surge in research and exploration focused on the utilization of causal learning methods to estimate brain effective connectivity. Generally, these methods can be categorized into two types: traditional machine learning methods and deep learning methods. Traditional machine learning methods [7,8,9,10,11,12,13] have the advantage of fewer parameters of model and are easy to interpret, which mainly include dynamic causal model (DCM) [14,15], structural equation model (SEM) approaches [16,17], Bayesian network (BN) methods [11,12], linear non-Gaussian acyclic model (LiNGAM) methods [8,18], Granger causality (GC) methods [19,20] and so on. However, such methods rely on assumptions about the model or data. With the rapid development of deep learning techniques, several deep learning methods have been applied exploratively to identify brain effect connectivity networks from fMRI data, such as multilayer perceptron (MLP) [21,22], recurrent neural networks (RNN) [23,24], and generative adversarial networks (GAN) [23,25]. These methods are more flexible and scalable, enabling the design of neural networks based on the characteristics of fMRI data. However, these methods are usually necessary to train models for each subject to estimate the brain effective connectivity network, which may lead to difficulties in fully utilizing the shared effective connectivity information between subjects and extremely high computational costs.

In this paper, we propose a novel framework for estimating effective connectivity based on an amortization transformer, named AT-EC. In detail, AT-EC first employs an amortization transformer encoder to estimate the brain EC across different subjects and a decoder to model the dynamics of fMRI time series under the learned brain EC network. This enables us to train an amortized model that leverages the shared knowledge from different subjects. Then, an assisted learning mechanism is designed that uses the statistical dependency information of FC networks to guide the estimation of EC networks. We have tested our model on both simulated and real-world fMRI data, and the experimental results show that the proposed method has certain advantages in performance compared with existing state-of-the-art methods.

The main contributions of this paper can be summarized as follows:We propose an end-to-end single amortization model that can model fMRI time series dynamics for different subjects and learn brain EC network. This model allows us to estimate EC with unseen subjects without refitting the model.We develop an assisted learning mechanism that uses the brain functional connectivity network as an additional embedding to guide the estimating of the brain EC network.Systematic experiments on both simulated and real-world data show that the proposed method achieves better performance compared to some state-of-the-art methods.

## 2. Related Work and Preliminary

This section presents a related work of brain effective connectivity estimation and provides the preliminary of amortization learning paradigm.

### 2.1. Brain Effective Connectivity Estimation

In this section, we introduce some state-of-the-art methods for estimating brain effective connectivity from fMRI data and summarize their advantages and disadvantages. We first introduce the traditional machine learning methods, i.e., Bayesian network (BN) methods and Granger causality (GC) methods, and then show the deep learning methods.

The Bayesian network (BN) method [26] employs probabilistic graphical models that can represent causal relationships between brain regions and has been widely used to infer EC from fMRI data. Bayesian networks provide a flexible and expandable framework for searching brain EC networks, but they are computationally demanding and cannot model cyclic or bidirectional EC.

The Granger causality (GC) method [27] calculates how past values of one brain region time series improve the prediction of another brain region time series, and estimate causal relationship (effective connectivity). These methods can learn brain EC networks from fMRI time series data by using regression models. However, the traditional GC method only considers the causal relations between two variables.

With the wide success of deep learning technology, several novel brain EC learning methods based on deep learning technology have been constantly presented. Compared to traditional machine learning methods, deep learning methods are better at handling highly noisy and non-linear data. In detail, Liu et al. [25] present a brain EC learning algorithm based on generative adversarial networks (EC-GAN). The method uses the generative adversarial process to obtain brain EC networks, where the generator uses structural equation models (SEM) to quantify causal relationships among brain regions and the discriminator distinguishes between the joint distribution of real and generated fMRI time series. Li et al. [28] design a CR-VAE method, which learns EC by an encoder and multi-head decoders under the general concept of Granger causality. Zou et al. [29] offer an EC estimation method based on a controllable variational autoencoder (CVAEEC). This method first employs an encoder to encode the fMRI time series input as latent variables and then uses SEM to learn brain EC network during the generation of the fMRI time series data.

Although these methods achieve favorable results, they fail to fully utilize the shared effective connectivity information between subjects. To more accurately estimate brain EC, it is necessary to extract and utilize the knowledge shared between different subjects.

### 2.2. Amortization Learning

For traditional optimization problems such that:(1)$$y^{★}\left(x\right)\in \underset{y}{\arg\min}f\left(y;x\right),$$
where *x* denotes the input context, *y* is the output we control, $y^{★}\left(x\right)$ denotes the optimal solution to the optimization problem, which reflects the similarities between problems across various contexts. It is not explicitly defined, but rather emerges from the optimization process and is usually considered unique. However, since the optimization problem (Equation 1) usually does not have a closed-form solution, it is usually necessary to use approximate numerical methods during iterative optimization to obtain an approximate optimal solution with extremely high computational cost, such as the directed acyclic graph searching problem  [30].

Amortized learning can amortize the cost of solving the optimization problems across many contexts to approximate the solution mapping [31], and has become a widely used modeling technique, particularly in scenarios that need to solve multiple instances of the same problem repeatedly. This approach enables the prediction of solutions to such problems, making use of the structural similarities that exist between them [32].

The solution for the amortization learning can be expressed as follows:(2)$$\hat{\Phi }=\underset{\Phi }{\arg\min}E\left[L\left(f,Y,X,p\left(x\right),{\hat{y}}_{\Phi }\right)\right],$$
where X describes the context space, Y describes the control space, *f* denotes optimization objectives, p(x) represents the distribution of data, and ${\hat{y}}_{\Phi }$ denotes the approximate optimal solution, i.e., the mapping from X to Y, Φ denotes the solution-related parameters, in the context of deep learning, Φ can represent the parameters of a neural network. L needs to be optimized under all the contexts involved in amortization.

## 3. Methodology

In this section, we present our proposed novel model, i.e., AT-EC, which can estimate brain effective connectivity from fMRI time series data. Specifically, we first give an overview of the proposed method, and then describe the details of the main components. Finally, we show the description of AT-EC.

### 3.1. Main Idea

The AT-EC consists of two components: (1) Amortization transformer for fMRI time series dynamics modeling; and (2) estimating EC with FC guidance. First, the amortization transformer takes advantage of shared dynamics among different fMRI subjects to improve the prediction of brain EC. In detail, the amortization transformer is an encoder–decoder model, the encoder extracts Granger causal relations of fMRI time series, and the decoder predicts the next time step of the fMRI time series data under the estimated Granger causal relationship (brain EC). The concept of amortized learning is utilized to train an amortized model on multiple fMRI subjects, enabling it to learn the brain EC of new unseen subjects. In this way, the smaller the difference between the generated fMRI time series and the original fMRI time series, the more accurate brain EC we can obtain. However, the brain EC network of each subject is different, and even a generalized amortization model has difficulty accurately learning all the brain effective connectivities. Then, to obtain an optimal brain EC network, the FC network of each subject is as embedded weight matrices to guide the estimating of brain EC networks. The structure of the proposed AT-EC is shown in Figure 1.

### 3.2. Amortization Transformer for fMRI Time Series Dynamics Modeling

#### 3.2.1. Amortization Transformer Architecture

In this section, we introduce the amortization transformer architecture. The amortization transformer component is an encoder–decoder model, which takes the real fMRI time series data as input and generates brain EC network and fMRI time series data for loss function. We use the idea of amortization learning to train an amortized encoder–decoder model. As the optimization problem described in (Equation 2), X denotes fMRI time series data of different subjects and Y denotes the generated fMRI time series data, p(x) represents the distribution of fMRI time series data. ${\hat{y}}_{\Phi }$ represents the brain EC network and Φ is the parameter of encoder–decoder model. In order to obtain the optimal brain EC network, the optimized loss function L needs to be minimized among all the subjects involved in the training.

Given the fMRI time series of *m* subjects with *n* brain regions $x_{i}$
(i=1,…,n) and *t* length, the input training data *X* can be represented as:(3)$$\begin{array}{c} X={\left(x_{1},x_{2},\ldots ,x_{n}\right)}^{⊤}\in R^{m\times n\times t}, \end{array}$$
where the *m* subjects may have different brain EC networks, but most ECs are the same. Amortization transformer can learn the knowledge shared between these subjects and does not need to retrain the model when faced with new unseen subjects. Next, we introduce the encoder–decoder components in detail.

#### 3.2.2. Transformer-Based Encoder

The amortization encoder utilized in this paper is based on the transformer model. When dealing with short-length fMRI time series, the transformer model can quickly capture important features of brain region series through multi-head attention mechanisms, and integrate these features into a global representation, thereby better extracting the causal relationships between brain regions. The transformer-based encoder involves first embedding the inputs via a linear layer, followed by processing them through multiple identical encoder blocks comprising a multi-head self-attention layer and a feed-forward layer; we posit that multi-head self-attention is well-suited for extracting temporal features from fMRI time series data, as it reduces reliance on external information and better captures internal correlations within the fMRI data. The encoder block operations are as follows:(4)$$\begin{array}{c} X^{\prime }=\mathrm{Linear}\left(X\right)\in R^{m\times n\times h},\\ Q_{l}=W^{Q}{X^{\prime }}^{l}+\epsilon^{Q},\\ K_{l}=W^{K}{X^{\prime }}^{l}+\epsilon^{K},\\ V_{l}=W^{V}{X^{\prime l}}^{l}+\epsilon^{V}, \end{array}$$
where $X^{\prime }$ denotes the embedded input, Linear is a fully connected linear layer that provides a linear transformation of the input $X^{\prime }$, *h* denotes the number of hidden layer nodes of the fully connected linear layer, and ${X^{\prime }}^{l}$ expresses the *l*-th input after dividing the embedding $X^{\prime }$ into L head self-attention layer. The attention mechanism can be described as a function that takes both a value and a set of key-query pairs as input and produces an output by calculating a weighted sum of the values. Q,K and *V* describe the query, key, and value, Q,K is used to calculate the similarity of the input features and acquire the attention features with value *V*. $W^{Q},W^{K}$ and $W^{V}$ denote the network parameters for the self-attention layer, respectively, $\epsilon^{Q},\epsilon^{K}$ and $\epsilon^{V}$ are the bias vector. Then, we can obtain $Q_{l},K_{l}$ and $V_{l}$, which denote the output of the linear transformation of the input ${X^{\prime }}^{l}$. So the self-attention can be calculated as follows:(5)$$\begin{array}{c} Attention_{l}=\mathrm{softmax}\left(\frac{Q_{l}K_{l}^{T}}{\sqrt{D_{K_{l}}}}\right)V_{l}, \end{array}$$
where $Q_{l},K_{l}$ and $V_{l}$ denote the query, key, and value vector of the self-attentive calculation of the *l*-th head, D denotes the number of elements in the last dimension of the query, key, and value vector $Q_{l},K_{l}$, and $V_{l}$, $Attention_{l}$ describes the *l*-th head attention vector. Here, the softmax function is used for stable training which has a lower computational cost than the other commonly used sigmoid function. Then, we can collect all *L* heads of the self-attention vectors as follows:(6)$$\begin{array}{c} X_{MultiHead}=\mathrm{Concat}\left(Attention_{1},Attention_{2},\ldots ,Attention_{L}\right). \end{array}$$

Finally, we can obtain the brain EC adjacency matrices G by putting $X_{MultiHead}$ into a feed-forward layer, which consists of two liner layers and a ReLU activation.
(7)$$\begin{array}{c} G\in R^{m\times n\times n}=\mathrm{softmax}\left(\mathrm{Feedforward}\left(X_{MultiHead}\right)\right), \end{array}$$
where *n* denotes the number of nodes (brain regions) and *m* is the number of fMRI subjects for amortization learning.

#### 3.2.3. MLP-Based Decoder

Based on the brain EC network G estimated by the amortization transformer encoder, we can define the decoder as follows:(8)$$\begin{array}{c} x_{n}^{t+1}=g\left(x_{n}^{\leq t},G\right)+\varepsilon_{n}^{t+1},t\in \left[1,T\right], \end{array}$$
where $\varepsilon_{n}^{t+1}$ is the exogenous noise variables, *t* describe the time step of the fMRI time series, *T* denotes the length of the time series. The decoder function *g* describes the dynamics of all time series data $x_{n}\in X$. In practice, the function *g* of the decoder can be defined as any model, such as a neural network model or predefined nonlinear formulation. Due to the high cost of data collection and pre-processing, the length of fMRI time series is usually short, so we consider a multilayer perceptron (MLP) with 3 hidden layers and a single activation function σ:R→R for the decoder:(9)$$\begin{array}{c} x_{n}^{t+1}=\mathrm{Linear}\left(x_{n}^{\leq t}\times G\right). \end{array}$$

MLP is a type of feed-forward neural network that performs well in short-term time series prediction tasks. Compared to the commonly used long short-term memory model (LSTM) and recurrent neural network (RNN), MLP has less parameters and is less prone to over-fitting. Therefore, using MLP may be a better choice.

The input of the decoder model is $x_{n}^{\leq t}$ and their brain EC network G. The purpose is to use the underlying causal relationships between brain regions to generate the time series of each brain region, which can be described as $x_{n}^{t+1}$. During the generation of fMRI time series data, the decoder strictly obeys the fundamental principles of Granger causality (GC), which results in a clear understanding of the data-generating process. When the generated fMRI time series data highly match the original fMRI time series data, we can obtain an optimal brain effective connectivity network. Therefore, the amortization transformer encoder and decoder parameters are trained with the reconstruction loss function.

#### 3.2.4. Loss Function

In this section, we present the loss function for the optimization task of AT-EC. The objective loss function is given such that:(10)$$\begin{array}{cc} f_{\phi_{★}},g_{\theta_{★}} & ={\mathrm{argmin}}_{f_{\phi },f_{\theta }}L\left(X,f_{\phi },g_{\theta }\right),\\ \;\mathrm{where}\;L\left(X,\phi ,\theta \right) & =\sum_{m=1}^{M}\sum_{t=1}^{T-1}\ell \left(X,g_{\theta }\left(X,f_{\phi }\left(X\right)\right)\right)+r\left(f_{\phi }\left(X\right)\right), \end{array}$$
where $f_{\phi }$ and $g_{\theta }$ denote the amortization transformer encoder and decoder, respectively, ϕ and θ are the parameters of them, *t* describes the time step of the fMRI time series, *m* is the number of the fMRI subjects for amortization learning, *X* denote the original fMRI time series inputs which collect all dimension in $R^{m\times n}$. *ℓ* denotes the metrics of data regression (i.e., evidence lower bound and least squares loss). In this paper, we use the evidence lower bound (ELBO) as the reconstruction loss function described as follows:(11)$$\begin{array}{c} \ell_{ELBO}=E_{f_{\phi }\left(G|X\right)}\left[\log g_{\theta }\left(X|G\right)\right]-\mathrm{KL}\left[f_{\phi }\left(G|X\right)∥p\left(G\right)\right]. \end{array}$$

The $\ell_{ELBO}$ loss is a negative log-likelihood value with a KL-Divergence to a prior distribution over G. When training the model, we want to maximize the log-likelihood and keep the KL-Divergence as small as possible. To overcome overfitting and infer sparse effective connectivity networks (causal graphs), we add an L1-regularization sparsity penalty $r\left(f_{\phi }\left(X\right)\right)={∥f_{\phi }\left(X\right)∥}_{1}$ in our loss function.

### 3.3. Estimating EC with FC Guidance

Existing research suggests that FC can explain the existence of connections between different brain regions, and EC can reveal the influence of these connections on behavioral and cognitive processes [2]. Typically, the study of brain functional connectivity provides the basis for the study of EC. To improve the accuracy of the brain EC network, we propose an assisted learning mechanism to estimate EC with FC guidance.

We first employ the Pearson correlation (PC), which is widely used to calculate the statistical dependencies of other variables, to obtain the skeleton of the causal diagram (FC network) of each subject [33]. Then, we use the FC network to guide the brain EC estimation. Next, we introduce the processes of PC and EC estimation in turn.

Assuming the brain FC network learned by PC is *W* and $w_{i,j}\in W$, where $w_{i,j}$ describes the statistical correlation of variables (brain region) between $x_{j}$ and $x_{j}$. $w_{i,j}$ is calculated as follows:(12)$$\begin{array}{c} w_{x_{i},x_{j}}=\frac{\mathrm{cov}(x_{i},x_{j})}{\sigma_{x_{i}}\sigma_{x_{j}}}, \end{array}$$
where cov(.) represents the covariance between two variables, and $\sigma_{x_{i}},\sigma_{x_{j}}$ is the standard deviation of the two variable $x_{i},x_{j}$. The value of $w_{i,j}$ and $w_{j,i}$ should be the same since the FC network has no direction and is not affected by changes in the position or size of the two variables $x_{i}$ and $x_{j}$.

Based on the learned FC network, we can modify the EC network similar to the attention mechanism, and we consider the FC network *W* as the query-key pair and the EC network learned by the amortization transformer as the value vector. Therefore, the estimating EC with FC Guidance can be obtained by:(13)$$\begin{array}{c} G^{\prime }=G\times \mathrm{Softmax}(W), \end{array}$$
where G∈G denotes the EC network of a single subject, which has the same dimension with FC network *W* in $R^{n\times n}$. Here, the softmax function is used to normalize the FC network *W*, and the properties of the softmax function can be used to highlight the weights of important brain regions to guide the learning of the EC network. $G^{\prime }$ is the final brain EC network of the subject.

### 3.4. Algorithm Description

Above all, the AT-EC algorithm mainly consists of two phases: fMRI time series dynamics modeling phase, and estimating EC with FC guidance phase. The algorithm is formally stated in Algorithm 1.
**Algorithm 1 **AT-EC
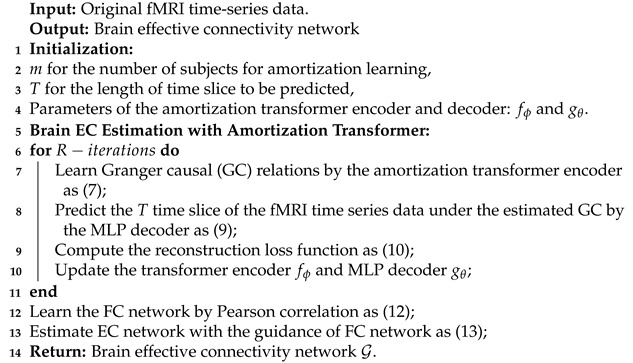


(1) In the fMRI time series dynamics modeling phase, AT-EC uses *m* fMRI subjects as an amortization learning task with simultaneous input models for training, then performs model initialization and sets some basic parameters. The fMRI time series of *m* subjects first enter the encoder simultaneously to obtain *m* brain EC networks. Then, based on these EC networks, *T* time slices are predicted by the MLP decoder. Finally, the predicted *T* time slices are used to calculate the reconstruction loss and guide the update of parameters. In each iteration, the learned EC network can be constantly optimized with the reduction of loss value. (2) In the estimating EC with FC guidance phase, the algorithm first use PC to learn the FC network for each subject, and then use the FC network to guide the brain EC estimation. Finally, we can obtain the optimal brain EC network for *m* subjects.

## 4. Experiment Setting

This section presents our experimental setting, which includes data description, baseline methods, evaluation metrics, and model configuration. To evaluate the performance of AT-EC, we compare AT-EC with other state-of-the-art methods using simulated fMRI data generated from known ground-truth networks to demonstrate its efficacy. Moreover, we also apply AT-EC to public real fMRI data to showcase its potential for practical applications.

### 4.1. Data Description

#### 4.1.1. Benchmark Simulation Data

The benchmark simulation data we used are supported by Smith et al. [7] and Sanchez-Romero et al. [34], which are generated by dynamic causal models [35] (DCM). Smith dataset (https://www.fmrib.ox.ac.uk/datasets/netsim/index.html) has been widely used in the literature owing to its rich, realistic simulated fMRI data for a wide range of underlying networks. In our experiment, we utilize three types of typical simulation cases to verify the performance of the AT-EC algorithm. In detail, sim1 is set to non-linearity. Sim2 shares the external inputs of a network. Sim3 adds the global mean confound to all nodes. All three datasets have 5 nodes, 50 subjects, 5 arcs, and 200 data points, which are instructive to simulate a real scene because the real scanning session time is usually shorter. The detailed information of the Smith benchmark simulation data is shown in Table 1.

The simple_network simulated fMRI data (https://github.com/cabal-cmu/feedback-discovery) is an expansion of the Smith dataset which increase in data points of the fMRI time series from 200 to 500 and reduces the original non-Gaussian of the BOLD data [36]. The ground-truth networks of simple_network simulated fMRI data contain different bidirectional structures, which are similar to the actual brain effective connectivity network. In detail, sim4 has the same ground-truth network of sim1, sim5 has more additional bidirectional structures, and sim6 has more nodes and edges. The detailed information of the simple_network benchmark simulation data is shown in Table 2.

#### 4.1.2. Real fMRI Dataset

The real-resting-state fMRI data of 23 human subjects are obtained from the MTlnet database (https://github.com/shahpreya/MTlnet). These real fMRI data are acquired at TR (Repetition Time) = 1 s, 7 min fMRI sessions for each subject, resulting in 421 data points of the fMRI time series. We consider the following seven regions of interest (ROIs) from the medial temporal lobe, which is referred to [37]. The detailed information of ROIs is shown in Table 3.

### 4.2. Evaluation Metrics

To evaluate the effectiveness of methods for learning effective connectivity, we employ the following three widely-used graph metrics to analyze the learned results: Precision, Recall, F1-measure (F1), and Structural Hamming distance (SHD).
(14)$$\begin{array}{c} Precision=\frac{CA}{TA}, \end{array}$$
(15)$$\begin{array}{c} Recall=\frac{CA}{\left|G\right|}, \end{array}$$
(16)$$\begin{array}{c} F1=\frac{2\times Precision\times Recall}{Precision+Recall}, \end{array}$$
(17)$$\begin{array}{c} SHD=EA+MA+RA, \end{array}$$
where extra arcs (EA), missing arcs (MA), reverse arcs (RA), correct arcs (CA), and total arcs (TA) are obtained from comparison of the learned results and ground-truth networks. And G is the arc set of ground-truth networks. |G| is the cardinal number of the set G.

### 4.3. Baseline Methods

To illustrate the competitiveness of the AT-EC algorithm, we compare AT-EC with the other seven algorithms. These comparison algorithms contain classical machine learning methods and some novel deep learning methods, which are isGC (2017), ACOCTE (2022), EC-GAN (2020), CR-VAE (2023), and CVAEEC (2021) algorithms. The parameter configurations of the corresponding methods are shown in Table 4.

### 4.4. Model Configuration

In our framework, our encoder construction first adopts a Transformer encoder module with 6 layers and uses the ReLU activation function. The number of hidden nodes is 2048 and the number of multi-head attention groups is 4. Two feed forward layers are used at the end of the encoder module to identify Granger causal relationships between brain regions. The decoder simply adopts the MLP layer decoder, which is divided into two layers, and each layer contains three fully connected linear layers. We use the Adam optimizer to optimize and fix the learning rate of 0.0025, and the learning rate of test time adaptation logic is 0.05. Sparse punishment can help us more effectively observe the correct and sparse EC network and our sparse punishment threshold is set at 0.3. For model training, we use five subjects for amortized learning for 200 epochs. When facing new subjects, we only iterate training 10 epochs to obtain an optimal brain EC network. For baseline methods, we run or train each subject using the parameters shown in Table 4 with the same sparse punishment threshold of 0.3.

## 5. Experimental Results

### 5.1. Results on Benchmark Simulated fMRI Dataset

To evaluate the performance of the learning methods, we apply five baseline methods and AT-EC to the benchmark simulation datasets, the Smith simulated datasets have 50 subjects and the simple_network simulated datasets have 60 subjects. we run the six methods on every subject to simulate a real scene. We report the mean μ and standard deviation σ of results across all subjects and assess the performance of these six methods based on Precision, Recall, F1, and SHD. A method is considered to perform well if it achieves higher precision, recall, and F1 scores, with lower SHD values.

Results on simulation datasets are shown in Table 5. The bold values indicate that the method achieved the best results. Sim2 has the same ground truth as Sim1, but the external neuronal noises are mixed into the nodes. From the results of Sim2, we can notice that traditional machine learning algorithms are more affected by external noise inputs, the precision, recall, and F1 values of the IsGC approach have declined somewhat. The ACOCTE approach is a score-based search algorithm that has stronger noise resistance. The deep learning methods are less affected, the EC-GAN, CR-VAE, CVAEEC methods achieve better performance despite the influence of noise. Moreover, compared with other algorithms, AT-EC still maintains its relative advantages over other algorithms.

Sim3 includes global mean confounding factors into the fMRI time series of brain regions, which are variables that are not measured or controlled in a causal relationship. Many causal discovery methods will identify spurious effective connectivity under the influence of confounding factors. The results of Sim3 indicate that most algorithms are robust to global confounding factors. All other methods outperform or equal the results of Sim1, except the isGC approach. AT-EC achieves the best recall and was second only to F1 with EC-GAN, which means AT-EC estimates almost all sides but identifies too many two-way connections.

Sim4 reduces the original non-Gaussian of the BOLD data. From the results of sim4, we can notice that all methods traditional machine learning methods show improved performance, however, the deep learning methods perform worse. Theoretically speaking, deep learning methods have stronger expressive power and adaptability, and they should achieve better performance. We believe that the threshold for the set sparse penalty is too small (0.3), making it difficult to filter out invalid edges. This is confirmed by the small value of precision and the recall of one in the results. Compared to other methods, the AT-EC method still achieves the best results in terms of precision, recall, SHD, and F1.

Sim5 has two additional bidirectional structures, which are similar to the real scene since the bidirectional connection structures are common in the brain effective connectivity network. From the results of sim5, we find that ACOCTE and CVAEEC have high precision and a low recall, which means both methods learn brain EC with high accuracy and rarely learn the wrong EC, but miss some ECs. Conversely, EC-GAN and CR-VAE achieve high recall and low precision, indicating that the ECs were basically learned but many are learned incorrectly, probably because these methods identify too many two-way connections. Our method has a high accuracy and recall values, indicating that the learned ECs have few errors and missing.

Sim6 has more nodes and edges, which also are similar to the real scene, since the complex human brain can be divided into many brain regions, usually multiple brain regions collaborate to complete an action. From the results of sim6, we notice that the performance of most methods is declining, while CVAEEC and AT-EC methods are performing well. This may be because, for most methods, it is difficult to learn causal relationships for short-term time series. Thanks to our amortization transformer architecture and FC-guided EC learning mechanism, our approach still achieves the best performance (F1).

Overall, deep learning methods are capable of extracting deep features from fMRI data, allowing for more accurate and precise results. AT-EC achieves better performance than state-of-the-art deep learning algorithms since it exploits correlation information across subjects.

### 5.2. Results on Real Resting-State fMRI Dataset

Unlike simulated data, it is not possible to evaluate the performance of causal search algorithms on real fMRI data using fully defined ground truth. Instead, our evaluation relies on partial knowledge about the structural connections between brain regions in the medial temporal lobe based on existing studies [34,38].

For the real fMRI data, we run AT-EC on every individual subject (each subject has 421 time points) for the seven medial temporal lobe regions of interest of the left and right hemispheres separately. We also do the same on each baseline algorithm for each hemisphere and for individual cases the list of directed edges and their frequency of appearance across the 23 individual subjects. The EC between two brain regions is estimated when we consider edges that appear in 40% of the 23 individual subjects. Figure 2 illustrates the EC networks estimated by AT-EC and the baseline methods from the left hemisphere medial temporal lobe and right hemisphere medial temporal lobe.

In Figure 2f. The effective connectivity networks of the left and right hemisphere medial temporal lobes are similar, but exhibit some differences. These differences are mainly caused by the connections of CA1, CA23DG, SUB, ERC, BA35,36, and PHC. Effective connectivities CA1→BA36, CA23DG→ERC, SUB→CA23DG, PHC→CA1, and SUB→BA36 are in the left hemisphere while BA36→ERC and PHC→ERC are in the right hemisphere.

Compared with the other methods, AT-EC learns the most correct edges (red line) with few missed edges (blue line) in both the left and right hemispheres. As is suggested by Lavenex and Amaral [38], the flow of information from the medial temporal lobe cortices (BA35,BA36,PHC) directly into the entorhinal cortex (ERC) and travel to CA23DG to CA1, this is the main pathway connecting the medial temporal lobe cortices with the hippocampus. We can find that those important effective connectivities (CA23DG↔CA1, CA1↔SUB, BA35↔BA36, PHC→SUB and CA23DG→SUB) are both learned in the left and right hemispheres, and other methods cannot identify them accurately. We also missed some important brain effective connectivities, such as the one-way connection between ERC and CA23DG (ERC→CA23DG), which is an important connection in the main pathway connecting the medial temporal lobe cortices with the hippocampus. Overall, the new method AT-EC performs better than the state-of-the-art methods and could provide a reliable perspective for the analysis of brain effective connectivity networks.

## 6. Conclusions

Estimating brain EC from fMRI time series data is still a challenging problem in the study of the brain connectome. In this paper, we propose a novel EC estimation method based on an amortization transformer, named AT-EC. AT-EC first employs an amortization transformer to model the dynamics of fMRI time series and infers brain EC across different subjects. Then, an assisted learning mechanism is designed to assist the estimation of the brain EC network. Experimental results on both synthetic and real datasets show that AT-EC performs well compared to the state-of-the-art methods, which shows that the amortization causal discovery approach has great development potential in brain EC estimation. In the future, we will extend this work to more large-scale brain EC networks.

## Figures and Tables

**Figure 1 brainsci-13-00995-f001:**
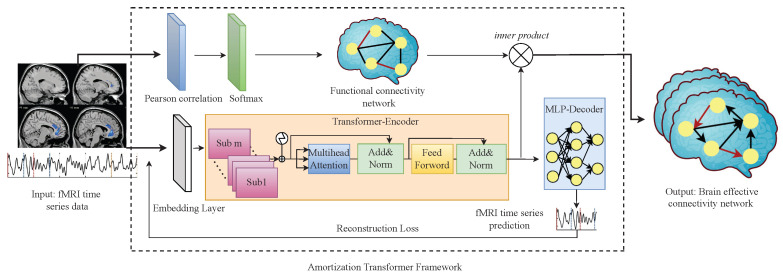
The structure of AT-EC.

**Figure 2 brainsci-13-00995-f002:**
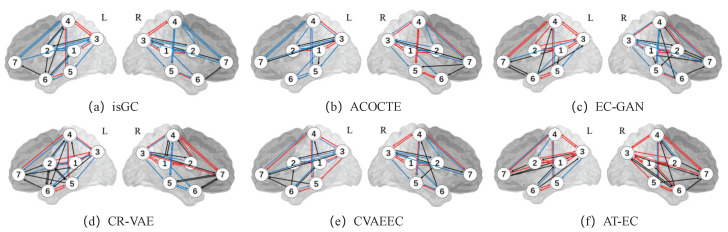
The EC network inferred by 6 methods on real fMRI data. The red lines represent the important connections between brain regions. In contrast, the black lines may indicate incorrect or spurious connections, while the blue lines may represent missing connections that were not captured by the algorithm. The ROIs include: (1) CA1; (2) CA23DG; (3) SUB; (4) ERC; (5) BA35; (6) BA36; and (7) PHC.

**Table 1 brainsci-13-00995-t001:** Description of the Smith benchmark simulation data.

Dataset	No. Arc	No. Node	Samples	Noise (%)	Other Factors	Session (min)	TR(s)
Sim1	5	5	200	1.0		10	3.00
Sim2	5	5	200	1.0	shared inputs	10	3.00
Sim3	5	5	200	1.0	global mean confound	10	3.00

**Table 2 brainsci-13-00995-t002:** Description of the simple_network benchmark simulation data.

Dataset	No. Arc	No. Node	Samples	No. Subject	Session (min)	TR(s)
sim4	5	5	500	60	10	1.20
sim5	5	7	500	60	10	1.20
sim6	10	18	500	60	10	1.20

**Table 3 brainsci-13-00995-t003:** The ROIs of the real fMRI dataset.

NO.	ROI	Detailed Description
1	CA1	Cornu Ammonis 1
2	CA23DG	Cornu Ammonis 2, 3 and Dentate Gyrus
3	SUB	Subiculum
4	ERC	Entorhinal Cortex
5	BA35	Brodmann Areas 35
6	BA36	Brodmann Areas 36
7	PHC	Parahippocampal Cortex

**Table 4 brainsci-13-00995-t004:** Parameter settings of these baseline methods.

Methods	Parameter Settings
isGC [27]	cmp = 4, ARorder = 2, normalize = 1
ACOCTE [26]	$\alpha =1.0,\beta =2.0,q_{0}=0.98,\rho =0.2$
EC-GAN [25]	lr = 0.01, dlr = 0.01, l1 = 5, II = 3, nh = 100, dnh = 100
CR-VAE [28]	context = 20, λ = 0.1, lr = 0.05, nh = 64
CVAEEC [29]	$\tau_{A}=0$, $\lambda_{A}=0$, $c_{A}=1$, lr = 0.003, nh = 64

**Table 5 brainsci-13-00995-t005:** The mean and the standard deviation results of 6 methods on benchmark simulated dataset using single subject data.

Data	Metrics	Methods					
		isGC	ACOCTE	EC-GAN	CR-VAE	CVAEEC	AT-EC
sim1	Precision	0.36 ± 0.20	0.44 ± 0.34	0.29 ± 0.04	0.22 ± 0.09	**0.60 ± 0.35**	0.52 + 0.01
	Recall	0.37 ± 0.21	0.27 ± 0.23	**0.93 ± 0.13**	0.41 ± 0.22	0.24 ± 0.15	0.80 + 0.02
	SHD	4.94 ± 1.65	3.80 ± 1.22	4.90 ± 0.85	6.32 ± 1.26	3.86 ± 0.74	**1.90 + 0.49**
	F1	0.35 ± 0.17	0.32 ± 0.26	0.44 ± 0.05	0.28 ± 0.12	0.32 ± 0.17	**0.62 + 0.01**
sim2	Precision	0.32 ± 0.20	0.45 ± 0.31	0.28 ± 0.04	0.26 ± 0.14	0.48 ± 0.24	**0.51 + 0.01**
	Recall	0.34 ± 0.22	0.33 ± 0.22	**0.91 ± 0.14**	0.48 ± 0.22	0.30 ± 0.16	0.82 + 0.02
	SHD	5.14 ± 1.45	4.00 ± 1.56	5.22 ± 0.84	6.10 ± 1.40	4.00 ± 1.07	**1.90 + 0.89**
	F1	0.31 ± 0.17	0.37 ± 0.25	0.43 ± 0.06	0.32 ± 0.14	0.35 ± 0.15	**0.61 + 0.00**
sim3	Precision	0.28 ± 0.15	0.52 ± 0.30	**0.66 ± 0.15**	0.24 ± 0.10	0.64 ± 0.30	0.47+0.01
	Recall	0.39 ± 0.21	0.42 ± 0.24	0.63 ± 0.25	0.41 ± 0.21	0.30 ± 0.13	**0.92 + 0.15**
	SHD	5.66 ± 1.61	3.20 ± 1.39	5.70 ± 1.43	6.14 ± 1.46	3.64 ± 0.73	**2.30 + 0.61**
	F1	0.32 ± 0.15	0.46 ± 0.26	**0.61 ± 0.11**	0.29 ± 0.13	0.39 ± 0.15	0.58 + 0.00
sim4	Precision	0.25 ± 0.00	**0.69 ± 0.21**	0.25 ± 0.00	0.25 ± 0.00	0.53 ± 0.19	0.64 ± 0.01
	Recall	**1.00 ± 0.00**	0.56 ± 0.18	**1.00 ± 0.00**	**1.00 ± 0.00**	0.30 ± 0.17	0.85 ± 0.01
	SHD	5.00 ± 0.00	2.62 ± 1.28	5.00 ± 0.00	4.98 ± 0.13	3.52 ± 0.89	**1.48 ± 0.78**
	F1	0.40 ± 0.00	0.62 ± 0.19	0.40 ± 0.00	0.40 ± 0.00	0.38 ± 0.15	**0.73 ± 0.01**
sim5	Precision	0.35 ± 0.00	**0.76 ± 0.17**	0.52 ± 0.00	0.35 ± 0.00	0.92 ± 0.12	0.66 ± 0.01
	Recall	**1.00 ± 0.00**	0.55 ± 0.12	**1.00 ± 0.00**	**1.00 ± 0.00**	0.54 ± 0.13	0.84 ± 0.01
	SHD	5.00 ± 0.00	3.12 ± 1.53	5.00 ± 0.00	5.00 ± 0.00	2.65 ± 0.84	**1.87 ± 0.72**
	F1	0.52 ± 0.00	0.64 ± 0.14	0.52 ± 0.00	0.52 ± 0.00	0.67 ± 0.11	**0.73 ± 0.01**
sim6	Precision	0.20 ± 0.00	0.53 ± 0.11	0.20 ± 0.00	0.33 ± 0.00	**0.86 ± 0.11**	0.59 ± 0.01
	Recall	**1.00 ± 0.00**	0.36 ± 0.07	**1.00 ± 0.00**	**1.00 ± 0.00**	0.42 ± 0.05	0.64 ± 0.01
	SHD	31.00 ± 0.00	11.00 ± 2.32	31.00 ± 0.00	31.00 ± 0.00	**9.97 ± 0.69**	10.23 ± 6.75
	F1	0.33 ± 0.00	0.42 ± 0.09	0.33 ± 0.00	0.33 ± 0.00	0.56 ± 0.05	**0.60 ± 0.01**

## Data Availability

The datasets used for this study are available in the original studies cited.

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
