# Peer review of "Amortization Transformer for Brain Effective Connectivity Estimation from fMRI Data"

_brainsci, 2023, doi:10.3390/brainsci13070995_

Round 1

Reviewer 1 Report

This research article focuses on the problem of inferring brain effective connectivity using fMRI data. The proposed method utilizes an amortization transformer to model fMRI time series data and introduces an auxiliary model training mechanism to enhance the effectiveness of the model. The experimental results demonstrate the efficacy of the proposed approach.

Strengths

1. The paper addresses a challenging and competitive topic in neuroscience, contributing to the field by tackling the problem of inferring brain effective connectivity from fMRI data.

2. The rationale for the approach is well-justified, and the paper provides comparisons with traditional machine learning approaches and deep learning methods, highlighting the advantages of the proposed approach.

3. The organization of the paper is clear and coherent, making it easy to understand. The proposed framework is illustrated in detail, allowing for straightforward comprehension and implementation.

Minor concerns

1. The meaning of QKV with subscript "l" in Equation 4 could be further clarified to enhance understanding.

2. The role of the softmax function in Equation 12 should be explained to provide a better understanding of its purpose and contribution to the model.

3. It would be beneficial to provide a definition and description of the ELBO (Evidence Lower Bound) to enhance clarity and comprehension. If necessary, including a formula would be helpful.

4. The symbol "X_i" on line 143 should be written as a bold lowercase letter to maintain consistency with the notation used in the paper.

5. The term "Subjects" should be in the singular form as "Subject" on line 316 to ensure grammatical accuracy.

The overall quality is good, but minor editing is suggested.

Reviewer 2 Report

1. Need to elaborate the abstract.

2. Introduction part was not sufficient.

3. Improve the literature survey of your study.

4. Explain the FMRI concepts in the study.

5. explain the need of two components in the AT-EC

6.  What is the purpose of MLP-Based Decoder?

7. Explain the need of Amortization Transformer for fMRI.

8.Benchmark Simulation Data was not sufficient for comparison.

9. Improve the experimental setup. Discuss all the experimental setup.

10. Comparison results to be discussed in detail.
